# Cross-Cultural Adaptation and Validation of Oral Health Literacy Instrument (OHLI) for Malaysian Adults

**DOI:** 10.3390/ijerph17155407

**Published:** 2020-07-28

**Authors:** Muhammad Zulkefli Ramlay, Norkhafizah Saddki, Mon Mon Tin-Oo, Wan Nor Arifin

**Affiliations:** 1School of Dental Sciences, Universiti Sains Malaysia, Kubang Kerian, Kelantan 16150, Malaysia; zull_um@yahoo.com; 2Biostatistics and Research Methodology Unit, School of Medical Sciences, Universiti Sains Malaysia, Kubang Kerian 16150, Malaysia; drmonmontinoo@gmail.com

**Keywords:** cross-cultural adaptation, oral health literacy instrument (OHLI), Malay version, validation studies

## Abstract

Currently, the availability of a functional oral health literacy instrument in the Malay language is limited. This study aimed to cross-culturally adapt Oral Health Literacy Instrument (OHLI) into the Malay language and to determine its psychometric properties in Malaysian adults. Cross-cultural adaptation of the OHLI into the Malay version (OHLI-M) was conducted according to a guideline, followed by a cross-sectional study among outpatients in a selected health clinic. The psychometric evaluations were the comparison of the OHLI-M scores by education levels and last dental visits, the correlation of the reading comprehension section of OHLI-M with the Malay version of the Short Test of Functional Health Literacy in Adults (S-TOFHLA-M), the correlation of OHLI-M with decayed, missing, and filled teeth (DMFT) and Community Periodontal Index (CPI), and the test-retest reliability of OHLI-M. A total of 195 outpatients participated in this study. The OHLI-M scores were significantly different between participants with different levels of education and timing since last dental visit. Participants with lower secondary school qualification and below, and those whose last dental visit was more than two years ago or never, had significantly lower OHLI-M scores. There was a positive correlation between the reading comprehension scores of the OHLI-M and the S-TOFHLA-M (Spearman’s rho = 0.37, *p* < 0.001). There was no significant correlation between the OHLI-M scores and the DMFT index scores or the CPI scores. The internal consistency was good (Cronbach’s alpha = 0.83 to 0.88). The test-retest reliability was excellent (intraclass correlation = 0.80 to 0.86). The OHLI-M showed good validity and reliability among adults in Malaysia.

## 1. Introduction

Health literacy is a relatively new concept in health promotion that has gained increasing importance in public health over the last two decades. There are various definitions of health literacy in the literature. The range of definitions reflects the complex and multidimensional nature of the construct [1]. The widely accepted definition states that “health literacy is the degree to which individuals have the capacity to obtain, process and understand basic health information and services needed to make appropriate health decisions” [2]. This definition denotes the acquisition of knowledge, skills, and confidence, which determine the motivation and ability of individuals to gain access, understand and use information in ways that improve, promote, and maintain good health [3].

Several instruments have been developed over the years to measure health literacy. The most widely used instruments are the Rapid Estimate of Adult Literacy in Medicine (REALM) [4] and the Test of Functional Health Literacy in Adults (TOFHLA) [5]. The REALM is a word recognition test that measures the ability of a person to correctly pronounce selected medical terms and lay terms for body parts and illnesses [4], and TOFHLA is a reading comprehension test that measures a person’s ability to understand and apply written information by evaluating both reading and numeracy skills [5].

Both REALM and TOFHLA are generic instruments that are designed to measure broad aspects of health literacy. Therefore, these instruments are potentially suitable for a wide range of patient groups and the general population. Oral health related terms are specific and different from the terms used in general medicine. In addition, there are many differences between general health and oral health, including disease progression and outcomes, clinical settings, treatment procedures, services and education messages, so that there is an indication of having specific instruments to measure a person’s ability to understand dental terms and oral health information [6].

Drawing on a broader understanding of health literacy, oral health literacy is defined as “the degree to which individuals have the capacity to obtain, process, and understand basic oral health information and services needed to make appropriate health decisions and act on them” [7]. Most oral health literacy instruments were developed based on the REALM or the TOFHLA [8]. Early oral health literacy instruments include the Rapid Estimate of Adult Literacy in Dentistry (REALD), which was adapted from the REALM [9,10], and the Test of Functional Health Literacy in Dentistry (TOFHLiD), which was modelled from the TOFHLA [11]. These instruments, however, received similar criticisms directed at the original health literacy instruments, in that they only measure word recognition, numeracy and reading skills in relation to oral health content, rather than oral health literacy per se [8].

Functional oral health literacy is required to understand and act on the information written on drug prescription labels, appointment cards, pre-operative and post-operative instructions, consent forms, educational materials, and other essential oral health-related materials [8]. In view of the issues with the existing instruments, the Oral Health Literacy Instrument (OHLI) was developed based on the TOFHLA to evaluate the functional oral health literacy in adults [12]. The OHLI contains both reading comprehension and numeracy sections to measure a person’s ability to perform oral health literacy tasks that require reading comprehension and numeracy skills. The psychometric properties for the entire OHLI questionnaire and its sections were found to be adequate [12].

Oral diseases pose a major health burden for many countries, including Malaysia. While there has been a downward trend in the prevalence of dental caries among adults in Malaysia, from 90.3% in 2000 to 88.9% in 2010, the prevalence remains considerably high [13]. Dental caries experience also does not seem to change. The average number of teeth affected by caries (decayed, missing, or filled) was 11.34 in 2000 and 11.66 in 2010 [13]. In addition, the prevalence of periodontal conditions in the adult population of Malaysia has been shown to be high and increasing. The proportion of dentate adults with healthy periodontium was only 3.2% in 2010 compared to 9.8% in 2000 [13]. Given the critical role of oral health literacy as a strong predictor of oral health disparities [14], the assessment of oral health literacy has gained considerable attention in recent years.

The availability of a functional oral health literacy instrument in the Malay language for the Malaysian population was limited. Hence, the objectives of this study were to conduct a cross-cultural adaptation of the OHLI into the Malay language and to determine its psychometric properties.

## 2. Materials and Methods

### 2.1. Study Setting and Participants

A cross-sectional study was conducted to determine the psychometric properties of the Malay version of the OHLI (OHLI-M) in Malaysian adults. For feasibility and logistic reasons, the sample was obtained from a health clinic in Seremban district of Negeri Sembilan state, Malaysia. Seremban district is the largest district in Negeri Sembilan. The chosen clinic provides a variety of health care services, which include out-patient services, antenatal and child health care, oral health care, physiotherapy, occupational health therapy, laboratory services, pharmacy services and emergency services. The clinic is located in a suburb, which covers the surrounding community from both urban and rural areas.

The inclusion criteria for participants were Malaysian citizens aged 18 years and above who could read, write, and understand the Malay language. The exclusion criteria were patients who came with a medical emergency and those who were mentally challenged.

Three different sample sizes were calculated based on different aspects of the planned psychometric assessments, which were the comparison of OHLI-M scores by education levels and last dental visits, the correlation of OHLI-M with decayed, missing, and filled teeth (DMFT) and Community Periodontal Index (CPI), and the test-retest reliability of OHLI-M by intraclass correlation coefficient (ICC). For the comparison of the mean OHLI-M scores between educations levels and last dental visits, the two-mean formula was applied, with alpha = 0.05, power = 80%, standard deviation = 18 [12], expected difference = 10 and expected drop-out rate = 20%, yielding *n* = 51 per group. For the comparison of three groups, the total sample size was 191 participants. For determining the correlation of OHLI-M with DMFT and CPI, the sample size was 85 participants at alpha = 0.05, power = 80% and medium effect size [15]. Lastly, the sample size for the ICC (test-retest reliability) was 49 participants by applying the formula [16] for two repetitions, with alpha = 0.05, power = 80%, lowest limit of acceptable ICC = 0.6, expected ICC = 0.8 and expected drop-out rate = 20%.

Potential participants were selected using systematic sampling. On average, it was estimated that 400 outpatients attend the clinic per day. Based on the planned data collection period, the time required to complete survey forms, and the researchers’ ability to handle no more than 20 participants per day, it was decided that a sampling interval of 20 was needed to sample about 20 outpatients per day. In addition, from this sample, every second participant was selected for further clinical examination. For the test-retest, a subsample of the participants was selected by convenience sampling to complete the OHLI-M again after two weeks. This was based on their willingness and cooperation to return to the clinic.

### 2.2. Translation and Adaptation of Oral Health Literacy Instrument (OHLI)

#### 2.2.1. English Version of OHLI

The English version of OHLI [12] has two sections: a reading comprehension section which assesses the ability to read and understand information related to oral diseases, and a numeracy section which evaluates the ability to understand instructions that require basic mathematical operations.

The reading comprehension section comprises of two passages, one on dental caries and the other on periodontal disease. The passage on dental caries contains 13 sentences with 264 words and 18 words omitted from the sentences. The passage on periodontal disease contains 14 sentences with 228 words and 20 omitted words. These omitted words serve as test items. In total, there are 38 items in the reading comprehension section. This section is self-administered. Respondents must choose a correct answer from four possible choices offered for each test item.

The numeracy section comprises of a series of printed prompts: five labels for medication labels frequently prescribed by dentists, a dental appointment card and a post-extraction instruction. In total, there are 19 test items in this section. The numeracy section is administered by face-to-face interview. The prompts are shown to the respondents and an interviewer will ask questions related to the prompts. The responses are recorded by the interviewer in a scoring sheet. Each correct answer receives a mark, and incorrect or no answer receives zero marks.

The final score for each section is the sum of all the items in the respective section. The total score for the reading comprehension section is multiplied by 1.316 (50/38) and the total score for the numeracy section is multiplied by 2.632 (50/19). This will give the weighted score for each section, ranging from 0 to 50. The sum of these weighted scores results is the total score for the OHLI, ranging from 0 to 100. The higher the OHLI score, the higher the functional oral health literacy. In addition, the OHLI score can be categorized into three oral health literacy levels: inadequate (0–59), marginal (60–74), and adequate (75–100).

#### 2.2.2. Malay Version of OHLI

The process of cross-cultural adaptation of the OHLI into Malay language followed the guideline by Beaton et al. [17] with several modifications to accommodate available resources. The process started with forward translation of the OHLI from English into the Malay language by two translators. Both translators were proficient in English and their native language was Malay. The first translator was a dental public health specialist, who was informed about the concepts assessed by the OHLI prior to the translation. The second translator did not have a dental background, representing a naïve translator who was blinded to the concepts assessed by the OHLI. Each translator worked independently.

Next, the two forward translations were reviewed by a review committee, comprising of two dental public health lecturers and two dental public health doctorate students. All members were fluent in both English and Malay languages. The committee synthesized the two forward translations to produce a combined forward translation of the OHLI. The combined forward translation of the OHLI was sent to two other translators who worked independently to translate the forward translation back into English. Both translators did not have dental background and were neither aware nor informed about the concepts assessed by the OHLI. Both translators were fluent in English and Malay. 

After the completion of backward translation, the review committee compared the backward translations against the original English version of the OHLI. The aim was to consolidate all forward and backward translations and to assess four aspects of equivalence, namely semantic equivalence, idiomatic equivalence, experiential equivalence, and conceptual equivalence. After reaching a consensus on the equivalence between the English and Malay versions, the pre-final Malay version of the OHLI (referred to as OHLI-M) was produced.

The OHLI-M retained the original structure of the English OHLI with an equivalent number of sentences and test items in reading comprehension passages. The number of prompts, types of prompts and the number of test items in the numeracy section were also equivalent to the English version. The OHLI-M was scored in the same way as the original OHLI.

The pre-final OHLI-M was pretested on a convenient sample of ten outpatients attending Hospital Universiti Sains Malaysia (HUSM) dental clinic in Kelantan, Malaysia. Participants who met the inclusion and exclusion criteria were invited to participate. At this stage, participants’ understanding of the items and the responses was assessed and any difficulties in interpreting the items were recorded. Comments and suggestions from the participants were evaluated by the review committee and necessary corrections were made accordingly. The review committee concluded that the changes required were minor and a second pretest sample for the OHLI-M was unnecessary.

The readability of OHLI-M was assessed using the Khadijah Rohani’s Readability Formula [18], which is the only formula currently available to assess readability of text in the Malay language. The formula is as follows:

Khadijah Rohani’s Readability Level = A − 13.988, where
$$A=\left(\frac{\mathrm{Number}\;\mathrm{of}\;\mathrm{words}}{\mathrm{Number}\;\mathrm{of}\;\mathrm{sentences}}\times 0.3793\right)+\left(\mathrm{Number}\;\mathrm{of}\;\mathrm{syllables}\times 0.0207\right)$$

The resulting value is interpreted based on the number of years of formal education according to the Malaysian education system [19]. The readability of passage 1 of the OHLI-M was level 10, which is equivalent to 4th grade in secondary school or to 10th grade in the American education system. The readability of passage 2 of the OHLI-M was level 5, which is equivalent to 5th grade in primary school or 5th grade in the American education system.

### 2.3. Additional Variables

#### 2.3.1. Socio-Demographic Characteristics

Information on the following socio-demographic characteristics of the participants was obtained: age, gender, race, education level and last dental visit. The education level categories used in a previous national survey among adults in Malaysia [13] were applied in this study: Level I for tertiary education (university), Level II for the equivalent of O-levels to institutions of higher learning (college, vocational/polytechnic institution, Malaysian Higher School Certificate or the equivalent, and Malaysian Certificate of Education or the equivalent), and Level III for lower secondary school qualification and below, including primary school and no formal education. The time since last dental visit was categorized into three groups: within the past 1 year, between 1 to 2 years and more than 2 years or never sought care.

#### 2.3.2. Malay Version of the Short Test of Functional Health Literacy in Adults (S-TOFHLA-M)

In addition to the OHLI-M, the reading comprehension of health literacy was measured by the Malay version of the Short Test of Functional Health Literacy in Adults (S-TOFHLA-M) [20]. The S-TOFHLA-M contains two passages with 36 omitted words. The first passage is about a healthy diet and the second passage is about a healthy lifestyle. The healthy diet passage contains eight sentences with 16 omitted words. The healthy lifestyle passage contains 11 sentences with 20 omitted words. For each omitted word, the respondent must select a word that best completes the sentence from a list of four possible words. Each correctly selected word is rewarded with one point. The raw score ranges from 0 to 36, which is multiplied by 2.778 (100/36) to create a total score between 0 to 100.

#### 2.3.3. Decayed, Missing, and Filled Teeth (DMFT)

Dental caries experience was measured by the DMFT index [21]. Each respondent was examined in supine position on a portable dental chair under an artificial light source with blue-white color spectrum. The assessment was performed using a disposable mouth mirror and an explorer. The numbers of decayed, missing, and filled teeth due to caries were recorded as D, M and F respectively, which range from 0 to 32 each.

#### 2.3.4. Community Periodontal Index (CPI)

Periodontal health status was measured by the CPI [21]. The assessment setting was similar to that of the DMFT index. The assessment was performed using the CPI probe with a 0.5 mm ball tip, a black band between 3.5 and 5.5 mm and rings at 8.5 mm and 11.5 mm from the ball tip. Scores were recorded according to the WHO guidelines [21].

### 2.4. Ethics

Permission to use the English version of OHLI [12] was obtained from the original authors and the study protocol was approved by the Universiti Sains Malaysia (USM) Human Research Ethics Committee Universiti Sains [USM/JEPeM/140378] and the Ministry of Health Medical Research and Ethics Committee [NMRR-13-1160-17164(IIR)]. Written informed consent was obtained from all patients who agreed to participate.

### 2.5. Statistical Analysis

Data were entered and analyzed using IBM SPSS Statistics version 22. Numerical and categorical socio-demographic variables were summarized as *n* (percent) and mean (standard deviation (SD)), respectively. Total scores of the OHLI-M and the S-TOFHLA-M were summarized and estimated by mean (SD) and 95% confidence intervals (CI).

For psychometric properties, the comparison of the OHLI-M scores by education levels and last dental visits was done by one-way ANOVA. Then, the correlation of the reading comprehension section of the OHLI-M with the S-TOFHLA was done by Spearman’s rank correlation, while the correlation of the OHLI-M with the DMFT and the CPI was done by Pearson’s correlation. The correlation coefficient values were interpreted according to the recommendation by Munro [22]: little if any correlation (0.00–0.25), low correlation (0.26–0.49), moderate correlation (0.50–0.69), high correlation (0.70–0.89), and very high correlation (0.90–1.00). Lastly, the reliability of the OHLI-M was evaluated by examining its internal consistency and its test-retest reliability by Cronbach’s alpha and ICC (two-way mixed, absolute agreement, single measure) respectively. Cronbach’s alpha values > 0.7 were considered as good internal consistency [23,24]. The ICC values were interpreted according to Cicchetti [25]: poor agreement (<0.4), fair agreement (0.40–0.59), good agreement (0.60–0.74), and excellent agreement (>0.74).

## 3. Results

The OHLI-M and the S-TOFHLA-M were tested on a sample of 195 outpatients from the selected clinic. After the initial completion of the OHLI-M and the S-TOFHLA-M, 85 participants underwent a dental clinical examination to determine their dental caries and periodontal disease status using the DMFT index and the CPI. In addition, 50 participants completed the OHLI-M after two weeks for the test-retest.

The socio-demographic characteristics of the participants are presented in Table 1. The mean age was 32.4 years (SD = 10.53). Most of the participants were female (56.5%) with a mean age of 32.4 years (SD = 10.53) and were predominantly Malay (78.5%). The highest education level for most participants was Level III (lower secondary school qualification and below including primary school and no formal education level). Most participants visited dentists in the past two years (62.0%). Descriptive statistics for the OHLI-M and the S-TOFHLA-M scores are presented in Table 2 as mean scores and 95% CIs.

The OHLI-M scores were significantly different between participants with different levels of education and time since last dental visit. Participants with Level III education (lower secondary school qualification and below) had significantly lower OHLI-M scores than participants with higher education (Level I and Level II). In addition, participants who had their last dental visit more than two years ago or never visited a dentist had significantly lower OHLI-M scores than participants who visited a dentist in the past one year (Table 3).

The correlation between the reading comprehension section of OHLI-M scores and S-TOFHLA-M scores was examined by Spearman’s rank correlation because the variables were not normally distributed. The correlation was positive (Spearman’s rho = 0.37, *p* < 0.001), in support of convergent validity between the OHLI-M and S-TOFHLA-M. However, there was no significant correlation between the OHLI-M scores and DMFT index or CPI scores (Table 4), which showed lack of support for concurrent validity. The internal consistency reliability and test-retest reliability of OHLI-M were good with Cronbach’s alpha of 0.83 to 0.88 and ICC of 0.80 to 0.86 (Table 5).

## 4. Discussion

The availability of the functional oral health literacy instrument in the Malay language was limited, thus the purpose of this study was to cross-culturally adapt the English version of the OHLI into the Malay language (OHLI-M) and to provide its psychometric properties. This was achieved by producing the OHLI-M, followed by validating the instrument in a sample of outpatients in a health clinic in Negeri Sembilan, Malaysia.

The validity of the cross-cultural adaption of the OHLI was ensured by a thorough process of translation, adaptation, and review in this study. To maintain the original structure of the OHLI, only minor changes to passages in the reading comprehension section were made to fit the Malaysian context. The changes were made to the answer choices by replacing the choices with conceptually similar words (e.g., *muffins* replaced with *kuih baulu* (baulu cake)), culturally less sensitive words (e.g., *drinking* replaced with *smoking*) and grammatically-correct words in the context of the sentence construction in the Malay language (e.g., *since* replaced with *meletakkan* (put)). In the numeracy section, the five prescription labels, dental appointment label and questions related to these labels from the original OHLI were kept, because the medicines and amount prescribed were similar to that of the prescription made by dentists in Malaysia. However, it was found that the post-operative instructions and three questions related to the instructions were not applicable in Malaysian oral health care settings. Thus, the post-operative instructions and the three questions were replaced by post-extraction instructions from MyHEALTH Portal of the Ministry of Health of Malaysia [26].

The OHLI-M was able to differentiate groups of participants with different education levels and time since the last dental visit. This supported the validity of the OHLI-M in relation to the observed groups, commonly referred to as known-group validity. This study showed a significant difference between education levels, with participants from Level III (lower secondary school and below) having the lowest mean OHLI-M score, while participants from Level I (university) had the highest mean score. This finding was consistent with the results from the English OHLI [12] and the Russian OHLI [27] which showed a significant difference in the OHLI scores by education level and higher OHLI scores for participants with higher education. This study also corroborated the results for both the English and Russian OHLI which showed significantly lower scores for participants with longer time since last dental visit.

The reading comprehension scores of the OHLI-M and the S-TOFHLA-M were positively correlated. This supported the convergent validity of the OHLI-M in relation to other established oral health literacy instrument. This was similar to the English OHLI which showed positive correlation of the reading comprehension scores between the OHLI and the TOFHLA [12]. However, unlike the study by Sabbahi et al. [12], the numeracy scores of the OHLI-M were not compared to that of the TOHFLA. This was because the S-TOFHLA-M that was utilized in this study lacked the numeracy section.

Oral health literacy is an important predictor of an individual’s oral health. In particular, studies have shown that lower oral health literacy is associated with poorer oral health outcomes [28,29]. A significant negative correlation was found between the Chilean OHLI scores and all dental clinical indices measured in the study [30]. However, in this study, the OHLI-M scores (reading comprehension, numeracy, and total) were not correlated with any of the clinical variables measured, which were the DMFT index scores and the CPI scores, indicating somewhat poor concurrent validity of the OHLI-M. Nevertheless, this issue deserves further study in the future because the correlation between the OHLI scores and the clinical variables such as the DMFT and the CPI was not studied in the English and Russian OHLI, so it is difficult to assert that the OHLI-M lacks concurrent validity.

The OHLI-M showed good internal consistency reliability and test-retest reliability. The Cronbach’s alpha values in this study were all more than 0.8, even after considering the lower 95% CI. The results for the reading comprehension section and the entire instrument were consistent with the English, Russian, and Chilean OHLI, all of which exceeded 0.8 [12,27,30]. However, the Cronbach’s alpha value in this study was higher (>0.8) than the English and Chilean OHLI for the numeracy section [12,30], while it was consistent with the Russian OHLI which also showed a Cronbach’s alpha value of more than 0.8 [27].

The test-retest reliability results showed excellent agreement for the reading comprehension, numeracy, and total scores. This indicates the temporal stability of the OHLI-M with repeated measurements on different occasions. The finding in this study further strengthens the test-retest reliability of the OHLI as indicated by the ICC results in the English, Russian, and Chilean OHLI studies, all of which showed excellent ICC values for the total OHLI scores [12,27,30]. However, the results varied for the reading comprehension and numeracy scores between these studies [12,27,30].

This study had a number of limitations that must be acknowledged. First, the members of the review committee were limited to dental public health professionals. Instead, it is recommended that at minimum, the committee must comprise of methodologists, health professionals, language professionals, and the translators (forward and back translators in the consolidation stage) [17]. Second, the pre-final OHLI-M was pretested among outpatients in a dental clinic in Kelantan instead of Negeri Sembilan, where the main validation study was conducted. There might be some differences in how the participants understood the items in the OHLI-M due to the difference in the Malay dialects used in Kelantan and Negeri Sembilan. However, despite this limitation, the standard Malay language used in the OHLI-M was understood by all participants. Third, there was a high refusal rate among those aged 50 and above, because the time spent to complete the survey forms could delay their dentist appointment. This could affect the validity of the OHLI-M in the older age group because of the low representativeness in this study sample. Lastly, for feasibility and logistic reasons, the sample was taken from a selected clinic in Negeri Sembilan. The sample in this study closely reflects the composition of the Malaysian population by gender and education level [31,32]. However, the Chinese were under-represented in this study compared to the composition of the Malaysian population by race (this study: 7.2%, Malaysia: 22.6%) [31]. This could affect the generalizability of this study to Malaysian Chinese.

## 5. Conclusions

The cross-culturally adapted OHLI-M showed good validity and reliability in adult outpatients in Malaysia. Based on the evidence, the OHLI-M can be used to assess functional oral health literacy of Malaysian adults. However, it is recommended that the OHLI-M is cross-validated in other Malaysian states in future studies to provide additional evidence of validity and reliability of the instrument.

## Figures and Tables

**Table 1 ijerph-17-05407-t001:** Sociodemographic characteristics of the respondents (*n* = 195).

Variables	*n* (%)
Age (years; mean [SD])	32.4 (10.53)
**Age (category)**	
<30	92 (47.2)
30–39	63 (32.3)
40–49	20 (10.3)
50–59	17 (8.7)
≥60	3 (1.5)
**Gender**	
Male	85 (43.6)
Female	110 (56.4)
**Race**	
Malay	153(78.5)
Chinese	14 (7.2)
Indian/Pakistani	27(13.8)
Other Bumiputera	1 (0.5)
**Education**	
Level I	49 (25.1)
Level II	59 (30.3)
Level III	87 (44.6)
**Last Dental Visit**	
<1 year	72 (36.9)
1–2 years	49 (25.1)
>2 years or never sought care	74 (37.9)

**Table 2 ijerph-17-05407-t002:** OHLI-M and S-TOFHLA-M scale level statistics (*n* = 195).

Scale	Mean (SD)	95% CI	Minimum	Maximum
OHLI-M				
Reading comprehension	37.6 (7.34)	36.5, 38.6	10.5	49.7
Numeracy	37.6 (11.02)	36.0, 39.5	0.0	50.0
Total	75.1 (15.64)	72.9, 77.4	15.8	97.4
S-TOFHLA-M	94.0 (11.81)	92.4, 95.7	30.5	100.0

Abbreviations: CI = confidence interval, OHLI-M = Malay version of Oral Health Literacy Instrument, SD = standard deviation, S-TOFHLA-M = Malay version of Short Test of Functional Health Literacy in Adults.

**Table 3 ijerph-17-05407-t003:** Comparison of total OHLI-M scores by education level and last dental visit (*n* = 195).

Variable	*n*	Mean (SD)	*F*-Statistic ^1^ (df1, df2)	*p*-Value
Education				
Level I	49	83.8 (10.10)	29.61 (2, 186)	<0.001 ^2^
Level II	59	79.1 (10.41)		
Level III	87	67.6 (17.64)		
Last dental visit				
<1 year	72	78.2 (15.20)	3.82 (2, 188.3)	0.020 ^3^
1–2 years	49	76.2 (11.40)		
>2 years/never sought care	74	71.5 (17.83)		

^1^ Brown-Forsythe modified *F*-test was used due to violation of equal variances assumption. ^2^ Post-hoc analysis with Dunnet’s T3 test shows significant difference between “Level I - Level II” and “Level II - Level III” education level pairs. ^3^ Post hoc analysis with Dunnett’s T3 test shows significant difference between “<1 year ago>2 years ago/never sought care” pair. Abbreviations: OHLI-M = Malay version of Oral Health Literacy Instrument, SD = standard deviation, df = degrees of freedom.

**Table 4 ijerph-17-05407-t004:** Correlation between OHLI-M, DMFT and community CPI for concurrent validity (*n* = 85).

Scale	DMFT	CPI
OHLI-M: Reading comprehension	*r* = 0.03 (*p* = 0.75)	*r* = −0.45 (*p* = 0.68)
OHLI-M: Numeracy	*r* = −0.17 (*p* = 0.10)	*r* = −0.03 (*p* = 0.81)
OHLI-M: Total	*r* = −0.11 (*p* = 0.33)	*r* = −0.04 (*p* = 0.70)

CPI, Community Periodontal Index; DMFT, Decayed, Missing, and Filled Teeth Index; OHLI-M, Malay version of Oral Health Literacy Instrument; *r*, Pearson’s correlation coefficient.

**Table 5 ijerph-17-05407-t005:** Internal consistency (by Cronbach’s alpha, *n* = 195) and test-retest reliability (by intraclass correlation, *n* = 50) of OHLI-M.

OHLI-M Scale	Cronbach’s Alpha (95% CI)	ICC (95% CI) ^1^
Reading comprehension	0.83 (0.80, 0.85)	0.84 (0.74, 0.91)
Numeracy	0.88 (0.85, 0.90)	0.80 (0.67, 0.88)
Total	0.88 (0.86, 0.89)	0.86 (0.72, 0.93)

^1^ Two-way mixed model, absolute agreement. Abbreviations: CI = confidence interval; ICC = intraclass correlation coefficient; OHLI-M, Malay version of Oral Health Literacy Instrument.

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
