# Peer review of "Cross-Cultural Adaptation and Validation of Oral Health Literacy Instrument (OHLI) for Malaysian Adults"

_ijerph, 2020, doi:10.3390/ijerph17155407_

Round 1

Reviewer 1 Report

Overall, a good study and manuscript. However, there are quite a number of grammatical errors, particularly in relation to the tense, in various segments of the manuscript - including at the beginning of the abstract - that need to be addressed.

Abstract: Correct grammatical errors.

Introduction: Good.

Methods: Line 91 - replace "handicapped" with “challenged”.

Results: Well described.

Discussion: A good discussion of the OHLI Scale.

Reviewer 2 Report

Thank you for disseminating this work. Please see the reviewers comments below as a guide to enhance this body of work.

Page 3 Line 105 regarding this statement: On average, 400 outpatients attended the clinic per day. Did you mean on average 400 participants visited the outpatient clinic?

Lines 106-108:  Please rephrase this statement and check spacing:

Considering the planned data collection period, the time required to complete survey forms and the ability of the researchers to handle at most 20 participants  per day, it was decided that a sampling interval of 20 was required to sample about 20 outpatients per  day.

Line 119-121 regarding this statement: The dental caries passage contains 13 sentences with 264 words and 18 words 120 omitted from the sentences, and the periodontal disease passage contains 14 sentences with 228 words and 20 omitted words.

Can you split the two categories for better clarity?

Backward translation was the best approach!

Page 4 Line 181 regarding the following statement please rephrase it: The readability of passage 1 and passage 2 is equivalent to secondary 4 and primary 5 respectively

Please clarify, is it 4th grade in secondary education and 5th grade in primary education? It is not very clear.

Lines 170-178, This section (below) could benefit from a chart or visual explanation or even sequential charting.

The readability of OHLI-M was assessed using Khadijah Rohani’s Readability Formula [16]. The 171 formula allows the assessment of readability of texts in Malay language. The assessment involved 172 performing the following mathematical operations: (1) Calculate the number of sentences in the 173 OHLI-M; (2) Divide 300 words with the number of sentences obtained from step (1) to give the 174 number of words per sentence; (3) Calculate the total number of syllables; (4) Multiply the value from 175 step (2) with 0.3793; (5) Multiply the value from step (3) with 0.0207; (6) Sum up the values from steps 176 (4) and (5); and (7) Subtract the value from (6) with 13.988. The resulting value from the step (7) is the 177 readability score of OHLI-M, which indicates the number of years of formal education required to 178 read the text according to the Malaysian education system [17].

Page 6 Lines 253-254: please rephrase the following statement or split it into two sentences.

Participants with lower secondary school qualification and below, and those whose last dental visit was more than two years ago or never visited dentist had significantly lower OHLI-M scores compared to their respective comparison groups (Table 3).

Table 3 is redundant; you can remove it.

Please edit the entire manuscript for space between sentences such as page 8, line 284 as …context.   The modifications were made to the answer choices by replacing the choices with conceptually similar words.

Reviewer 3 Report

I really enjoyed reading the paper as oral health literacy if of great public health significance and interest to me. There are some suggestions to improve the paper.

  1. English language and referencing corrections throughout the paper are. required.
  2. Introduction - Some background on oral health burden in Malaysia is required? How is health literacy a determinant of poor oral health needs to be articulated?
  3. Methods - Why was Khadijah Rohani’s Readability Index used compared to other indices? Did the authors trial the readability with other indices?
  4. Methods - Does the chosen study population and setting bias the results? How is the study population and demographic compared to the general population? More educated? Higher SES? Rural vs urban? This can impact health literacy.
